# Divalent Cation Modulation of Ion Permeation in TMEM16 Proteins

**DOI:** 10.3390/ijms22042209

**Published:** 2021-02-23

**Authors:** Dung M. Nguyen, Hwoi Chan Kwon, Tsung-Yu Chen

**Affiliations:** 1Center for Neuroscience, University of California, Davis, CA 95616, USA; dmanguyen@ucdavis.edu (D.M.N.); hwckwon@ucdavis.edu (H.C.K.); 2Pharmacology and Toxicology Graduate Program, University of California, Davis, CA 95616, USA; 3Biophysics Graduate Program, University of California, Davis, CA 95616, USA; 4Department of Neurology, University of California, Davis, CA 95616, USA

**Keywords:** TMEM16A, TMEM16F, phospholipids, divalent cations, permeability ratio

## Abstract

Intracellular divalent cations control the molecular function of transmembrane protein 16 (TMEM16) family members. Both anion channels (such as TMEM16A) and phospholipid scramblases (such as TMEM16F) in this family are activated by intracellular Ca^2+^ in the low µM range. In addition, intracellular Ca^2+^ or Co^2+^ at mM concentrations have been shown to further potentiate the saturated Ca^2+^-activated current of TMEM16A. In this study, we found that all alkaline earth divalent cations in mM concentrations can generate similar potentiation effects in TMEM16A when applied intracellularly, and that manipulations thought to deplete membrane phospholipids weaken the effect. In comparison, mM concentrations of divalent cations minimally potentiate the current of TMEM16F but significantly change its cation/anion selectivity. We suggest that divalent cations may increase local concentrations of permeant ions via a change in pore electrostatic potential, possibly acting through phospholipid head groups in or near the pore. Monovalent cations appear to exert a similar effect, although with a much lower affinity. Our findings resolve controversies regarding the ion selectivity of TMEM16 proteins. The physiological role of this mechanism, however, remains elusive because of the nearly constant high cation concentrations in cytosols.

## 1. Introduction

The TMEM16 family encompasses transmembrane proteins functioning as Ca^2+^-sensitive Cl^-^ channels and phospholipid scramblases [1,2]. TMEM16A is a Ca^2+^-activated Cl^-^ channel [3,4,5], and its activation is important for anion transport across cell membranes in many types of cells [6,7]. On the other hand, upon activation by Ca^2+^, TMEM16F and other fungal TMEM16 molecules scramble membrane phospholipids, as well as conducting ionic currents [8,9,10,11]. Recent structural-functional studies of TMEM16 molecules have revealed that the homodimeric architecture is conserved between anion channels and phospholipid scramblases, and helices 3–8 appear to form a conduit thought to be the pathway for ion and/or phospholipid transport [12,13,14,15,16,17]. However, the substrate transport pathways of the two types of TMEM16 phospholipid scramblases may be slightly different. In fungal phospholipid scramblases, the entire conduit appears as a groove with an open sidewall [12,13]. This conduit in mammalian TMEM16F, however, has an intact protein sidewall at its extracellular half, as shown in recent cryo-EM studies [17,18]. The difference in the degree of sidewall opening of this conduit between fungus and mammalian phospholipid scramblases raises a question regarding the molecular mechanism by which these scramblases transport phospholipids [18]. The ion-conducting pore of TMEM16A is similar to that of mammalian TMEM16F; namely, the pore is fully enclosed for the extracellular half but is missing part of the protein sidewall at the intracellular half [14,15,16]. As TMEM16A and TMEM16F are membrane proteins, the open cavity at the intracellular end of the conduit could consist of membrane phospholipids. Indeed, interactions of phospholipids and TMEM16 proteins have been suggested in structural and calculation studies [13,18,19,20,21,22]. Functional effects of phospholipids on the activation and rundown of TMEM16 molecules have also been experimentally documented [22,23,24,25].

Various studies generally agree that after activation by intracellular Ca^2+^, TMEM16A conducts mostly anions [5,26,27,28,29,30]. On the other hand, the cation versus anion selectivity for TMEM16F’s current conduction has been controversial. In TMEM16F, some experiments showed that TMEM16F non-selectively conducts cations and anions with only a slightly higher selectivity towards anions [31,32,33], whereas in other studies the selectivity was found to significantly favor cations [10]. This controversy may depend on experimental conditions. For example, experiments using whole-cell recording methods tend to report that TMEM16F is less cation-selective [31,32,33], whereas experiments with excised inside-out patch recordings have shown very different Na^+^ versus Cl^-^ permeability ratios (P_Na_/P_Cl_) [10,34]. Recently, Ye et al. [35] reported that the P_Na_/P_Cl_ ratio of TMEM16F depends on the intracellular Ca^2+^ concentration ([Ca^2+^]_i_) used to activate the current: the ratio was ~0.5 (more anion selective) and ~6–7 (more cation selective) when the TMEM16F current was activated by 1 mM and 15 µM [Ca^2+^]_i_, respectively. It was suggested that this effect could result from a change in the pore electrostatic potential due to Ca^2+^ binding to the pore. Our laboratory has shown that the electrostatic control of Cl^-^ flux through the TMEM16A pore can be achieved via altering the sidechain charge of a pore residue [34,36]. The binding of Ca^2+^ to the high-affinity channel activation sites has also been shown to electrostatically affect the ion conduction of TMEM16A [37], presumably because Ca^2+^-activation sites are near the intracellular pore entrance. The electrostatic control of Ca^2+^ in altering the P_Na_/P_Cl_ ratio of TMEM16F is thus conceivable, although the nature of this Ca^2+^-modulation effect on the P_Na_/P_Cl_ ratio in TMEM16F is still murky. For example, if TMEM16F activated by 15 µM and by 1 mM [Ca^2+^]_i_ have different P_Na_/P_Cl_ ratios, the difference cannot be mediated by Ca^2+^ binding to the activation sites of TMEM16F, which have an apparent Ca^2+^ affinity < 10 µM. Meanwhile, though Ye et al. [35] showed that TMEM16F was quite cation-selective (P_Na_/P_Cl_ ~ 6–7) when the current was activated by low [Ca^2+^]_i_ (15 µM), various studies in the literature revealed that [Ca^2+^]_i_ at similarly low concentrations activates a non-selective TMEM16F current [31,32,33,34].

We have reported that a high concentration (hundreds of µM or above) of intracellular Co^2+^ ([Co^2+^]_i_) potentiates the TMEM16A current induced by saturating [Ca^2+^]_i_ [38]. This Co^2+^ potentiation effect (see Figure 1A) appeared to be similar to the current potentiation by mM [Ca^2+^]_i_ after the activation of TMEM16A is saturated (see Figure 1B). We suggested that Ca^2+^ and Co^2+^ may bind to the pore region of TMEM16A to potentiate Cl^-^ currents [38]. In the present study, we further examined this current potentiation mechanism. We found that besides Ca^2+^ and Co^2+^, alkaline earth divalent cations such as Mg^2+^, Sr^2+^ and Ba^2+^ can all potentiate the TMEM16A current with a roughly similar low affinity. In addition, treating the patch intracellularly with poly-l-lysine reduces the potentiation effect, whereas lowering ionic strength enhances the potentiation. On the other hand, the same high concentrations of divalent cations had a minimal effect in potentiating the current of TMEM16F, but changed its P_Na_/P_Cl_ ratio, as reported by Ye et al. [35]. The P_Na_/P_Cl_ ratio of TMEM16F also depends on the concentration of intracellular monovalent cations. These results show a modulation on the ion permeation of TMEM16 molecules by a rather non-specific cation binding. We suggest that divalent and monovalent cations congregate around phospholipid head groups near or within the ion-transport pathway of TMEM16 proteins with low affinities, thus changing the electrostatic potential near the pore region. These observations are significant because they explain the controversy in the literature regarding the cation versus anion selectivity of TMEM16 molecules.

## 2. Results

We have previously shown that intracellular Co^2+^ (up to 20 mM) by itself does not activate current in WT_16A_. However, intracellular Co^2+^ potentiates the Ca^2+^-induced WT_16A_ current, followed by an inhibition effect [38]. Co^2+^ inhibition on the WT_16A_ current (see Figure 1A) is easier to understand—Co^2+^ competes with intracellular Ca^2+^ for the high-affinity Ca^2+^-activation sites [38]. The current potentiation by Co^2+^, however, is a different mechanism because the potentiation occurs before the inhibition. A high [Ca^2+^]_i_ also potentiates the current of WT_16A_ (Figure 1B), just like the potentiation generated by high [Co^2+^]_i_. For both Co^2+^ and Ca^2+^, at least hundreds of µM are required for generating a detectable potentiation effect. As the potentiation appears almost immediately upon the application of Co^2+^ or Ca^2+^, the effect likely results from a potentiation of Cl^-^ conduction through the TMEM16A pore [38].

To further explore the mechanism underlying Co^2+^ and Ca^2+^ potentiation, we examined whether other divalent cations also potentiate TMEM16A current. Figure 2A compares the dose-dependent potentiation of Mg^2+^ (upper panels) and Ca^2+^ (lower panels) on the WT_16A_ current. In these experiments, the current was induced by 300 µM [Ca^2+^]_i_ at +20 mV (colored in orange) and −20 mV (colored in blue), followed by the application of the indicated concentrations of extra divalent cations (2 mM to 20 mM). To compare the potentiation effects among different experiments, all currents were normalized to the control current (I_0_) right before the addition of the potentiating divalent cations. The degree of current potentiation, I_peak_/I_0_, was calculated, and was plotted against the concentration of the potentiating agents (Mg^2+^ or Ca^2+^). As shown in the right panels of Figure 2A, the current potentiation by Mg^2+^ and Ca^2+^ was qualitatively similar, both in the concentration range required for generating the effect, as well as in the degree of the potentiation. Figure 2B depicts the degree of potentiation of each individual patch (circle symbols) and the averaged potentiation (columns) by various divalent cations at 20 mM. The results show that alkaline earth divalent cations all potentiate the current of WT_16A_ to a roughly similar degree. An approximate 40%–60% increase at +20 mV and a 20%–40% increase at −20 mV of the control current (activated by 0.3 mM [Ca^2+^]_i_) can be further induced by 20 mM divalent cations.

The above results indicate that these various divalent cations potentiate the current of WT_16A_ with low affinities—all require at least hundreds of µM to mM to start generating the effect. This is different from the higher and very distinct apparent affinities of these divalent cations in activating the current of WT_16A_ [26]. The low affinity suggests that the potentiation might involve phospholipids because the binding affinities between the divalent cations and phospholipids are known to be in the range of hundreds of µM to mM, or above [39,40,41]. To explore whether the potentiation mechanism might involve phospholipids, we took advantage of the report that poly-l-lysine depletes membrane phospholipids [23]. Figure 3A,B illustrate Co^2+^ and Ca^2+^ potentiation, respectively, before and after poly-l-lysine treatment of the membrane patch expressing WT_16A_, or two Y589 mutants of TMEM16A with larger Co^2+^ potentiation, Y589V and Y589A (abbreviated as Y589V_16A_ and Y589A_16A_, respectively) [38]. Figure 3C,D show that exposing the membrane patches to intracellular poly-l-lysine (0.3 mg/mL) for only 5 s reduces the potentiation of Co^2+^ and Ca^2+^. On the other hand, after poly-l-lysine treatment, supplying the membrane patch with 20 µM L-α-phosphatidylinositol-4,5-bisphosphate (PIP2) for 1 min appears to recover the loss of potentiation caused by poly-l-lysine (Figure 3E,F).

The results shown in Figure 2B and in Figure 3C,D,F reveal a significant variation in the degree of potentiation among different membrane patches even before the treatment with poly-l-lysine. TMEM16 currents are known to undergo significant rundown, and poly-l-lysine facilitates the rundown process, an effect attributed to a depletion of phospholipids [23,42]. We thus examined if the degree of the potentiation correlates with channel rundown. Figure 4A shows an experiment on a TMEM16A mutant, Y589A_16A_. Here, the Mg^2+^ potentiation was tested every one minute at +20 mV and −20 mV. The rundown of the TMEM16A current can be directly observed from the time-dependent reduction of I_0_ from the recording traces in Figure 4A and from the averaged I_0_ in Figure 4B (circles). Figure 4B also shows that the averaged degree of the Mg^2+^ potentiation (I_peak_/I_0_, squares) was reduced during the current rundown. We also tested if mutations that reduce the binding of phosphatidylinositol-(4,5)-biphosphate (PIP2) to TMEM16A [22,42] alter the degree of potentiation. Figure 4C,D confirm that the current potentiation by 20 mM [Mg^2+^]_i_ is significantly smaller in P566A_16A_ and D481A_16A_ mutants than that in WT_16A_ whether the effects were examined at +20 mV or −20 mV. On the other hand, the mutation effects of K678Q_16A_ and R437Q_16A_ appear to be less pronounced—the potentiation in these two mutants is more similar to that in WT_16A_. Only the potentiation at −20 mV in R437Q is significantly smaller than that in WT_16A_. We speculated that the variation of these mutation effects in different mutants might be related to the physical distance of the mutated residues from the pore (see Discussion).

One well-documented mechanism underlying the effect of divalent cation binding to membrane phospholipids on ion channels is an alteration of the surface potential of the membrane [43,44]. Altering membrane surface potential affects the voltage-dependent activation of voltage-gated channels [45,46], as well as changing the conductance of various ion channels [47,48]. In altering the channel conductance, the effect depends on the distance of the surface charge from the pore due to the Columbic potential—the closer the charge to the pore, the larger the effect. A large groove cavity is found at the intracellular pore entrance of TMEM16 molecules, and various studies have suggested the presence of phospholipids in the nearby region [13,18,19,20,21]. As the open groove is exposed to phospholipids, divalent cations may bind to low-affinity binding sites formed by phospholipid head-groups near the pore, thus affecting ion transport. In comparison, the ion-transport pathways of CLC channel/transporters are completely enclosed by protein structures [49,50]. Therefore, membrane phospholipids are likely located further away from the pore entrance in CLC channels than in TMEM16A. We thus examined the degree of current potentiation by intracellular Mg^2+^ on a gate-opened mutant of CLC-0, a Cl^−^ channel expressed in a *Torpedo* electric organ. Indeed, the potentiation effects of intracellular Mg^2+^ in E166A_CLC0_ were much smaller than those observed in WT_16A_ (Appendix A), consistent with the idea that phospholipids are located closer to the ion transport pathway in TMEM16 molecules than in CLC channels. If the nature of the current potentiation is electrostatic, another prediction is that the potentiation will be enhanced in low ionic strength solutions [43,44]. Figure 5 shows that in symmetrical 40 mM [Cl^−^], the degrees of potentiation at +20 mV (outward current or inward Cl^−^ flux) by Co^2+^ and Mg^2+^ were greatly enhanced compared to those in the symmetrical 140 mM [Cl^−^] condition. However, lowering the ionic strength appears to have little effect on the potentiation of the inward current at −20 mV (see Discussion for the interpretation of these results).

The electrostatic control of channel conductance is thought to result from an increase in the local concentration of the counter-charged ions or a decrease in the concentration of the ions with the same charge near the pore [43,44,47]. Because TMEM16F conducts both cations and anions, we expected that the current potentiation in TMEM16F by divalent cations would be smaller than that in TMEM16A because TMEM16F conducts cations as well as anions. To study current potentiation in TMEM16F, we performed experiments on the Q559W mutant of TMEM16F (abbreviated as Q559W_16F_) due to the technical advantage of a slower current rundown in this mutant [34]. The recording traces shown in Figure 6A and the averaged data in Figure 6B indeed indicate that 20 mM Mg^2+^ or Co^2+^ generates very little current potentiation in Q559W_16F_. On the other hand, Mg^2+^ and Co^2+^ inhibit the Q559W_16F_ current (Figure 6A,C), likely by competing with Ca^2+^ for the high-affinity activation sites as that shown in TMEM16A [26,38].

The electrostatic potential that increases local [Cl^-^]_i_ and reduces local [Na^+^]_i_ near the pore entrance could alter the calculated Na^+^ versus Cl^-^ permeability ratio (P_Na_/P_Cl_). Controlling the channel’s ion selectivity by surface potential has been previously suggested from experimental data [51], as well as by theoretical analyses [52]. Recently, altering the P_Na_/P_Cl_ ratio of TMEM16F by intracellular Ca^2+^ has been reported [35]. To study the effects of divalent cations on the ion selectivity of TMEM16F, we employed Q559W_16F_ again because a more linear I–V curve in this mutant provides a better tool than WT_16F_ for measuring the reversal potential under asymmetrical ionic conditions [34]. The I–V curves of WT_16A_ and Q559W_16F_ shown in Figure 7A,B, respectively, were obtained from currents activated by 20 µM or 1 mM [Ca^2+^]_i_, using a voltage ramp from −80 mV to +80 mV. Compared to the symmetrical 140 mM [NaCl] condition, the reversal potentials with 40 mM or 15 mM [NaCl]_i_ were shifted towards the negative voltage direction for WT_16A_, but were shifted towards the positive voltage direction for Q559W_16F_, indicating that WT_16A_ is more Cl^—^selective, whereas Q559W_16F_ is more Na^+^-selective. Interestingly, the calculated P_Na_/P_Cl_ ratio in Q559W_16F_ showed a large difference between the currents activated by 20 µM [Ca^2+^]_i_ and those activated by 1 mM [Ca^2+^]_i_—the P_Na_/P_Cl_ ratios for the currents activated by 20 µM [Ca^2+^]_i_ were ~5.5–6.2 versus ~1.3–1.5 for the current activated by 1 mM [Ca^2+^]_i_ (Figure 7C). In WT_16A_, the P_Na_/P_Cl_ ratios between currents induced by 20 µM and 1 mM [Ca^2+^]_i_ appeared to be different as well, although the difference was much smaller (Figure 7C & Table 1). A significant difference of the P_Na_/P_Cl_ ratio between low and high [Ca^2+^]_i_ conditions is consistent with the previous finding of Ye et al. [35].

Motivated by the effects of intracellular poly-l-lysine and PIP2 on the divalent-cation activated TMEM16A current (Figure 3), we also examined whether the same reagents could alter the P_Na_/P_Cl_ ratio of Q559W_16F_. Figure 8A,B show that treating the intracellular side of the membrane patches with poly-l-lysine (0.3 mg/mL for 5 s) reduces the P_Na_/P_Cl_ ratio of Q559W_16F_, whereas PIP2 (20 µM for 1 min) can reverse the effect of poly-l-lysine. These results demonstrate that the current of TMEM16A and the P_Na_/P_Cl_ ratio in TMEM16F are both modulated by membrane phospholipid manipulation.

Because the degree of current potentiation of TMEM16A induced by divalent cations (such as Mg^2+^) is roughly similar to that induced by Ca^2+^, we asked whether intracellular Mg^2+^ can also change the P_Na_/P_Cl_ ratio of Q559W_16F_. Figure 9A,B show experiments in which 20 µM of [Ca^2+^]_i_ was used to activate the current. In an asymmetric Cl^-^ condition in which [NaCl]_o_ = 140 mM and [NaCl]_i_ = 40 mM, the P_Na_/P_Cl_ ratio of Q559W_16F_ was 4.9 ± 0.2 in the absence of Mg^2+^ but was reduced to 2.3 ± 0.1 in the presence of 1 mM [Mg^2+^]_i_, a result roughly similar to the effect of 1 mM [Ca^2+^]_i_. These results demonstrate that divalent cations other than Ca^2+^ can also alter the P_Na_/P_Cl_ ratio of Q559W_16F_.

In measuring the P_Na_/P_Cl_ ratios of Q559W_16F_ with asymmetric [NaCl], we noticed that the P_Na_/P_Cl_ ratio appears to be affected by [NaCl]_i_, especially in the experiments with low [Ca^2+^]_i_ (20 µM). For example, with 15 mM [NaCl]_i_, the current induced by 20 µM [Ca^2+^]_i_ has a P_Na_/P_Cl_ ratio of 6.2 ± 0.2, which is significantly larger than the P_Na_/P_Cl_ ratio of 5.5 ± 0.4 for the current induced by the same [Ca^2+^]_i_ in 40 mM [NaCl]_i_ (Table 1). Since the controversial P_Na_/P_Cl_ ratios from different laboratories were obtained with different intracellular concentrations of monovalent ions [10,31,32,33,34,35], we examined the effect of [NaCl]_i_ on the P_Na_/P_Cl_ ratio more rigorously. In this experiment, we used a non-charged molecule, mannitol, to replace [NaCl]_i_, and systemically changed [NaCl]_i_ in intracellular solutions containing 20 µM [Ca^2+^]_i_ (Figure 10A). The P_Na_/P_Cl_ ratios of the Q559W_16F_ current at 70 mM and 280 mM of [NaCl]_i_ were reduced to 2.8 ± 0.3 and 1.7 ± 0.1, respectively, in the same 20 µM [Ca^2+^]_i_ (Table 1). Figure 10B,C show that the ratio of P_Na_/P_Cl_ of Q559W_16F_ reduced with an increase of [NaCl]_i_, suggesting a reduction of cation selectivity in high [NaCl]_i_. It should be emphasized that except for the indicated [NaCl]_i_ the intracellular solutions in these experiments did not include other salts (mannitol was used to replace the reduced [NaCl]_i_). Although the affinities of phospholipid binding with monovalent cations are lower than those with divalent cations, the monovalent cation Na^+^ is likely the culprit, rendering the pore of Q559W_16F_ less cation-selective.

The above results are again consistent with the idea that cations bind to membrane phospholipids, alter pore potential and thus change the concentrations of permeant ions near the pore. This electrostatic control exists in both TMEM16A and TMEM16F. For TMEM16F, a change in the local ion concentration results in an alteration of the P_Na_/P_Cl_ ratio. However, since TMEM16A is much more selective for anions, the effect manifests as a potentiation of the anion flux through the pore. The phospholipids could reside within the pore, or at locations outside the pore within a distance short enough for an electrostatic effect (such as a position confined by helices 3, 4 and 5; see Figure 11), or both. The fact that the P_Na_/P_Cl_ ratio depends on divalent and monovalent cations and that the affinities of these cations to generate the effects are low (mM or above) further suggest that the binding of these cations is rather non-specific.

## 3. Discussion

Ion channels reside in lipid membranes, and thus membrane phospholipids can control the function of ion channels in multiple ways. PIP2, for example, directly interacts with various ion channels to regulate their functions [53]. The stabilization of the activity of voltage-gated Na^+^ channels by extracellular divalent cations was attributed to a membrane surface potential effect resulting from divalent cations binding to membrane phospholipids [45,46,54]. It has also been well documented that membrane surface potential can control the conductance of ion channels via altering the local ion concentrations near the cell membrane [43,47,48,55]. For TMEM16 proteins, depletion of PIP2 from the recorded membrane patches has been shown to reduce the apparent Ca^2+^ affinity for the activation of TMEM16F [23]. PIP2 also affects the rundown of TMEM16A and TMEM16F, perhaps by binding to domains formed by helices 3–5, as well as other parts of the molecules [22,23,24,42]. These examples underlie the critical interaction between membrane phospholipids and channel proteins. Here, we suggest that phospholipids that are tightly associated with TMEM16 molecules may participate in forming cation-binding sites in or near the intracellular ion transport pathway and that the binding of intracellular cations to the phospholipids alters pore potential and thus modulates ion permeation.

### 3.1. Divalent Cations Potentiate the Current of TMEM16A

All divalent cations and even the monovalent cations tested in the present study modulate the ion permeation of TMEM16 proteins at low affinities. In TMEM16A, generating the current potentiation requires at least hundreds of µM to mM Co^2+^ and alkaline earth divalent cations (Figure 1, also see [38]). The apparent affinity for monovalent cations (such as Na^+^) to modulate the ion permeation of TMEM16 molecules appears to be even lower, based on the observation that the calculated P_Na_/P_Cl_ ratios of TMEM16F at 15–40 mM [NaCl]_i_ are significantly different from those obtained at 70–280 mM [NaCl]_i_ (Figure 10). These very low affinities are similar to the binding affinities between phospholipids and various cations documented in the literature [39,41]. We thus suspected that divalent cations may bind to membrane phospholipids and mediate the effects by altering the membrane surface potential [43,44,56]. To examine whether the potentiation of TMEM16A current by divalent cations involves phospholipids, we conducted several lines of experiments. We first found that treating excised membrane patches with intracellularly applied poly-l-lysine for only 5 s reduced the potentiation of TMEM16A by divalent cations, whereas applying PIP2 can reverse the effect of poly-l-lysine (Figure 3). Secondly, the rundown of the TMEM16A current, a phenomenon thought to be due to phospholipid depletion, was correlated with a reduction in the current potentiation (Figure 4A,B). Finally, mutations thought to decrease PIP2 binding to TMEM16A also reduce divalent cation-induced potentiation (Figure 4D). These results support the idea that phospholipids are involved in the divalent cation-mediated potentiation of the TMEM16A current. It is possible that phospholipids are located at positions close to the pore of TMEM16A, and thus the binding of divalent cations can attract more Cl^-^ to the intracellular pore entrance. A depletion of negatively charged phospholipids (for example, PIP2) would thus result in less divalent cation binding to the pore region and weaken the current potentiation.

Because divalent cation-mediated potentiation appears to involve phospholipids, we further examined if the control is electrostatic in nature. Our results revealed that the potentiation of the outward TMEM16A current was significantly enhanced in solutions with a lower ionic strength (Figure 5), a phenomenon consistent with an electrostatic mechanism. We also compared the degree of potentiation between TMEM16A and a CLC channel because of the difference of the pore structures between these two types of Cl^-^ channels (Appendix A). Indeed, the same concentration of intracellular Mg^2+^ had a significantly smaller potentiation effect in a CLC channel compared to that in TMEM16A (Appendix A), a phenomenon that is likely due to the closer location of phospholipids to the intracellular pore vestibule of TMEM16A. It is interesting that the various mutations thought to alter the interaction of PIP2 with TMEM16 molecules reduce the Co^2+^-induced potentiation with different strengths (Figure 4D). Based on the cartoon in Figure 11B (right), where the locations of these four mutated residues relative to the pore vestibule are depicted, the strengths of the mutation effects may depend on the distance of these residues to the intracellular pore entrance. For example, mutations of D481 and P556 are known to affect the PIP2 located within a cavity formed by helices 3, 4 and 5 [42], a location quite close to the intracellular pore vestibule. Therefore, mutations reducing the amount of phospholipids at these positions are expected to exert a larger effect than the mutations located farther away (such as R437 and K678). Although other speculations may also exist, an electrostatic mechanism is consistent with the observation that larger effects occur at those two mutations closer to the pore vestibule.

### 3.2. Intracellular Cations Alter the Ion Selectivity of TMEM16F

If the TMEM16A current potentiation is mediated by a change of the potential near or within the pore, binding of divalent cations to the pore region is expected to cause an increase of local [Cl^-^]_i_ and a decrease of local [Na^+^]_i_. In an ion transport pathway that conducts both cations and anions (such as TMEM16F), an increase in Cl^-^ conductance would be canceled out by a decrease in Na^+^ conductance. Our experiments indeed show that the current potentiation by high concentrations (20 mM) of divalent cations is much smaller in Q559W_16F_ (Figure 6A,B). Nonetheless, the changes in local anions and cations are expected to alter the calculated cation versus anion selectivity, as shown elegantly in a theoretical analysis [52]. Our experimental results confirm that the cation selectivity of Q559W_16F_ in 1 mM [Ca^2+^]_i_ (P_Na_/P_Cl_ = ~1.3–1.5) was significantly smaller than that in 20 µM [Ca^2+^]_i_ (P_Na_/P_Cl_ = ~5–6) (Figure 7 and Table 1). Such effects of Ca^2+^ on the P_Na_/P_Cl_ ratio of Q559W_16F_ are consistent with those observed in WT_16F_ reported by Ye et al. [35].

Monovalent ions appeared to modulate the ion selectivity of TMEM16 proteins as well, as shown by the dependence of the P_Na_/P_Cl_ ratio of Q559W_16F_ on [NaCl]_i_ (Figure 10 and Table 1). With [Ca^2+^]_i_ = 20 µM, the P_Na_/P_Cl_ ratios of Q559W_16F_ were ~5–6 in 15–40 mM [NaCl]_i_. When [NaCl]_i_ was increased to 70–280 mM, the calculated P_Na_/P_Cl_ ratio was reduced to ~2. The binding affinities of phospholipids with monovalent cations have been shown to be lower than those with divalent cations and could be in the range of tens to hundreds of mM [39,41]. The observation that the P_Na_/P_Cl_ ratios at 15 and 40 mM [NaCl]_i_ were more similar to each other than to those at 70 mM [NaCl]_i_ is consistent with such a very low binding affinity between phospholipids and monovalent cations.

### 3.3. Modulations of TMEM16A and TMEM16F Ion Permeations Result from the Same Mechanism

Although divalent cations significantly alter the P_Na_/P_Cl_ ratio of TMEM16F, their effects on TMEM16A are manifested as an increase in the Cl^-^ current because Na^+^ flux through TMEM16A is nearly negligible. The degree of potentiation of the WT_16A_ current by divalent cations was larger at +20 mV (inward Cl^−^ flux) than at −20 mV (outward Cl^−^ flux). Since it was the intracellular divalent cations that were manipulated, the change in the surface charge should have been more prominent on the intracellular side of the membranes. A weaker potentiation of the inward current (outward Cl^-^ flux) therefore contradicts the classical Gouy–Chapman theory [43]. However, a more realistic surface potential model normally requires taking the location of the surface charge into account, as elaborated by the Stern theory [43,44,47,57]. The anomaly of a larger potentiation at +20 mV, for example, could result from the voltage-dependent binding of these modulating divalent cations if the binding sites are located within the membrane electric field. Interestingly, the effect of [Ca^2+^]_i_ on the P_Na_/P_Cl_ ratio of TMEM16F was also shown to be voltage dependent—in the same [Ca^2+^]_i_, the TMEM16F pore was more selective for anions at more positive membrane voltages [35]. In the present study (Figure 5), the potentiation of TMEM16A current was enhanced at +20 mV but not at −20 mV in symmetrical 40 mM [NaCl], a phenomenon that can also be explained by the idea that monovalent cations (such as Na^+^) modulate the pore potential of TMEM16A in a voltage-dependent way.

Although the divalent cation effects on altering TMEM16F’s P_Na_/P_Cl_ ratio have been reported by Ye et al. [35], our results here provide a more general insight underlying the modulation of the ion permeation of TMEM16 molecules. In both TMEM16A and TMEM16F, we observed several common features. First, we observed that all divalent cations and even monovalent cations can exert the effect with very low affinities, suggesting a non-specific binding. Second, we showed that phospholipids are involved in the modulation mechanisms. Third, the degree of modulation is different in solutions with different ionic strengths. These results led us to conclude that the potentiation of the TMEM16A current and the alteration of P_Na_/P_Cl_ ratio of TMEM16F result from the same electrostatic mechanism—intracellular divalent and monovalent cations likely bind to membrane phospholipids, thus modulating ion fluxes by altering the pore potential.

### 3.4. Implication of the Modulation of TMEM16 Current by Intracellular Cations

The insights from the present study led us to recover a coherent picture from the apparently controversial results of TMEM16F’s ion permeability in the literature—experiments with high concentrations of cations (including monovalent cations) on the intracellular side all showed a small P_Na_/P_Cl_ ratio (less cation selective). In fact, this phenomenon is not limited to TMEM16F, but appears to be also present in TMEM16A. When using 20 µM [Ca^2+^]_i_ to activate the current of WT_16A_, for example, the reversal potentials in solutions containing only 40 mM [NaCl]_i_ in this study (~−26 mV, see blue open squares in Figure 7A) were significantly smaller than those values (−30 to −31 mV) when the reduced [NaCl]_i_ was replaced with NMDG-Cl [36]. Thus, even in a quite anion-selective channel such as WT_16A_, if non-permeant monovalent cations (such as NMDG^+^) are used to replace the reduced [Na^+^]_i_, the estimated P_Na_/P_Cl_ ratio is slightly smaller than those from experiments where non-charged molecules (such as D-mannitol) were used to replace monovalent cations. This insight thus explains why TMEM16F is very cation-selective (P_Na_/P_Cl_ = 6–7) in excised membrane patch experiments in very low [NaCl]_i_ (for example, 15 mM) than in whole-cell recording experiments where physiological cation concentrations were present inside the cell.

It should be emphasized that although divalent cations exert a powerful effect on the calculated permeability on TMEM16F, a modulation resulting from altering the pore potential does not change the “intrinsic selectivity” of the channel pore [52]. The electrostatic control of the permeability ratio of two ions will occur if the two ionic species are of different valences. For two permeant ions with opposite charges traversing the pore (such as that in TMEM16F), the calculated permeability can be changed from more cation-selective to more anion-selective by means of a change in surface potential of tens of mV [52]. Such a degree of membrane surface potential perturbation is easily attainable by adding mM divalent cations [46,57,58].

Our studies thus clarify the biophysical mechanism of the divalent cation modulation on the ion permeation of TMEM16 molecules. The results, however, raise a question regarding the physiological role of this modulation. In vitro experiments show that the TMEM16F current can have a P_Na_/P_Cl_ = 6–7 at low [NaCl]_i_ (such as 15 mM). This high P_Na_/P_Cl_ ratio only exists if no other monovalent cations are present in the intracellular solution. In a physiological context, however, a high concentration of monovalent cations (such as K^+^) is present in cytosols. At the same time, the cytosolic solution also contains mM Mg^2+^ and other cations with multiple charges. As shown in Table 1, even with 70 mM intracellular monovalent cations, the P_Na_/P_Cl_ ratio is already reduced to ~2 in the absence of [Mg^2+^]_i_. A physiological concentration (>100 mM) of intracellular monovalent cations and mM [Mg^2+^]_i_ likely maintains a steady surface potential that renders TMEM16F non-selective even with a fluctuation of [Ca^2+^]_i_ from the sub-µM to µM level. What then is a possible physiological role for this modulation of ion permeation function of TMEM16 molecules? We speculate that the significance of this electrostatic control may lie in the metabolism of negatively charged phospholipids. For example, activation of G-protein-coupled receptors is known to stimulate phospholipase C activity, leading to the degradation of PIP2 [53]. Such a reduction in the amount of PIP2 in the inner leaflet of cell membranes might alter the electrostatic potential near the pore entrance of TMEM16 molecules and thus modulate ion permeation. It will require experiments in more physiological conditions to confirm or refute this possibility of controlling the ion permeation through TMEM16 molecules via synthesis and degradation of membrane phospholipids.

## 4. Materials and Methods

### 4.1. Molecular Biology and Channel Expression

The cDNAs of the TMEM16 family members and their mutants were generated as described recently in Nguyen et al. [38]. Briefly, the wild-type TMEM16A (WT_16A_) (NCBI reference sequence: NM_001242349.1) [59] and the wild-type TMEM16F (WT_16F_) cDNA (Addgene plasmid # 62554) were subcloned in pEGFP-N3 or pIRES2 expression vectors (Clontech/Takara Bio, Mountain View, California, USA). Mutants of TMEM16 molecules were created using the QuikChange II site-directed mutagenesis kit (Agilent Technologies, Santa Clara, California, USA) and the mutations were verified with commercial DNA sequencing services. The E166A mutant of CLC-0 (abbreviated as E166A_CLC0_), which was reported previously [60], was constructed in the pIRES2 vector (Clontech/Takara Bio, Mountain View, California, USA). Transfections of cDNAs to human embryonic kidney (HEK) 293 cells were performed using the Lipofectamine 3000 kit (MilliporeSigma, St. Louis, Missouri, USA) according to the manufacturer’s instructions. The cells that expressed channels were identified using green fluorescence with an inverted microscope (DM IRB; Leica) equipped with a fluorescent light source and a GFP filter (Chroma Technology, Bellows Falls, Vermont, USA).

### 4.2. Electrophysiological Recordings

All electrophysiological recordings were performed on inside-out membrane patches excised from GFP-positive cells 24–72 h after transfections. Except where indicated, the pipette (extracellular) solution was the same as the intracellular zero-Ca^2+^ solution, which contained 140 mM NaCl, 10 mM HEPES and 0.1 mM EGTA at pH 7.4 (adjusted with NaOH). Solutions with 20 µM Ca^2+^ were the same as the intracellular zero-Ca^2+^ solution except for the addition of 120 µM CaCl_2_ (pH adjusted to 7.4 after the addition of CaCl_2_). The experiments for evaluating the degree of current potentiation were performed in symmetrical [Cl^-^] (140 mM or 40 mM). For experiments in symmetrical 140 mM [Cl^-^] with intracellular solutions containing high concentrations (>20 µM) of divalent ions (Ca^2+^, Mg^2+^, Ba^2+^, or Co^2+^), EGTA was not included in the solution, and the total concentration of the divalent cations added to the solution was considered the free concentration. Because these divalent cations were from chloride salts, the amount of NaCl was reduced accordingly to maintain a total [Cl^−^] of 140 mM. For experiments testing the potentiation in symmetrical 40 mM Cl^−^, the extracellular (pipette) solution contains 40 mM [NaCl] and 100 mM D-mannitol (VWR Chemicals, Radnor, Pennsylvania, USA). The intracellular solution was of two kinds, containing 100 mM D-mannitol and 40 mM [NaCl]_i_ or 20 mM divalent cations (chloride salts). For experiments evaluating the reversal potential of TMEM16F, the extracellular solution was the zero-Ca^2+^ solution (140 mM [NaCl]). For measuring reversal potentials in asymmetrical [NaCl] conditions, if [NaCl]_i_ was lower than 140 mM (such as 15–70 mM), D-mannitol was used to replace the reduced [NaCl]_i_. All experiments were performed at room temperature (20 °C–22 °C). Poly-l-lysine-hydrochloride (MW 15000–30000, MilliporeSigma, St. Louis, Missouri, USA) was dissolved in ddH_2_O to make 150–300 mg/mL stock solutions, which were kept at −20 °C and were diluted into working solutions right before use. The intracellular side of the membrane patch was treated with poly-l-lysine (0.3 mg/mL) for 5 s. PIP2 of porcine brain origin (ammonium salt) was purchased from Avanti Polar Lipids (Alabaster, Alabama, USA). Upon use, the organic solvent in the vial (20:9:1, chloroform:methanol:water) was evaporated with nitrogen gas and immediately dissolved into the zero-Ca^2+^ working solution. The PIP2 solution (20 µM) was applied to the intracellular side of the membrane patch for 1 min.

Recording electrodes were made from borosilicate glass capillaries (World Precision Instruments) using the PP830 electrode puller (Narishige, Amityville, New York, USA). When filled with the zero-Ca^2+^ solution, the resistance of the electrodes was between ∼1.5 and ∼2.5 MΩ. All experiments were performed using the Axopatch 200B amplifier and the Digidata1440 analogue-digital signal-converting board controlled using pClamp10 software (Molecular Devices, San Jose, California, USA). Solutions were delivered to the intracellular side of the excised membrane patch using the SF-77 solution exchanger (Warner Instruments, Holliston, Massachusetts, USA). In all experiments, the ground electrode was immersed in 3-M KCl solution well, which was then connected to the bath solution with a 1% agarose salt bridge made out of 3-M KCl. Giga-ohm seal formations and patch excisions were always achieved in an identical bath and pipette solution. Liquid junction potentials were not corrected with the assumption that the potential difference among various intracellular solutions connected to 3-M KCl reference remain constant during recordings.

Except where indicated, all experiments of divalent cation-induced potentiation were initiated by clamping the membrane voltage to ±20 mV in the EGTA-containing zero-Ca^2+^ solution. The currents of TMEM16A or TMEM16F were activated by 0.3 mM [Ca^2+^]_i_. Intracellular divalent cations of various concentrations were then applied (in the presence of [Ca^2+^]_i_ used for channel activation) until the recorded current reached a steady state. The potentiating divalent cations were then removed (see Figure 1). Before analyzing the experimental results, the leak current in the absence of [Ca^2+^]_i_ was first subtracted from the recorded currents. For evaluating the degree of potentiation, the peak current in the presence of potentiating agents (I_peak_) was divided by the control current (I_0_) immediately before adding the potentiating agents (see Figure 1). In some experiments, the degree of current inhibition by divalent cations was also evaluated. In this case, the quasi-steady-state current at the end of the application of divalent cations was defined as I_Divalent_ (for example, I_Co_ or I_Mg_) (see Figure 1), and the ratio of I_Divalent_/I_peak_ was used to reflect the degree of inhibition. Data points from the dose-dependent divalent cation potentiation were not fitted to any binding curves because the potentiation by the highest concentration (20 mM) of divalent cations did not reach saturation. To estimate the Na^+^ and Cl^−^ permeability ratio (P_Na_/P_Cl_), the asymmetrical [NaCl] conditions were created by changing [NaCl]_i_ to 15, 40, 70 or 280 mM, while [NaCl]_o_ was kept at 140 mM. When [NaCl]_i_ was reduced, the reduced [NaCl]_i_ was replaced with the same concentration of D-mannitol (VWR Chemicals, Radnor, Pennsylvania, USA). The current-voltage (I–V) relationship was generated using a 1.6-s ramp protocol from −80 mV to +80 mV. For each patch, the leak current obtained in Ca^2+^ free solutions was subtracted from the current obtained in the presence of Ca^2+^, and the resulting leak-subtracted I–V curve was used to determine the reversal potential (E_rev_). I–V curves under symmetrical 140 mM [NaCl] were always recorded before [NaCl]_i_ was changed. If the reversal potential in symmetrical [NaCl] was more than 2 mV away from 0 mV, the patch was discarded. The P_Na_/P_Cl_ ratio was calculated from E_rev_ according to the Goldman–Hodgkin–Katz equation:(1)$$E_{\mathrm{rev}}=\frac{\mathrm{RT}}{F}\ln\frac{P_{\mathrm{Na}}{\left[{\mathrm{Na}}^{+}\right]}_{o}\;+{\;P}_{\mathrm{Cl}}{\left[{\mathrm{Cl}}^{-}\right]}_{i}}{P_{\mathrm{Na}}{\left[{\mathrm{Na}}^{+}\right]}_{i}\;+{\;P}_{\mathrm{Cl}}{\left[{\mathrm{Cl}}^{-}\right]}_{o}}$$
where R, T and F are the universal gas constant, absolute temperature and Faraday constant, respectively.

For the recording of E166A_CLC-0_ (Appendix A), because this mutant is open constantly, the tight seal of the excised patch was evaluated by taking advantage of the fact that the current of E166A_CLC-0_ was blocked by µM para-chlorophenoxy acetate (CPA) at negative membrane potentials [60]. Thus, for every excised patch, the currents at −160 mV in the absence and presence of 10 mM CPA (MilliporeSigma, St. Louis, Missouri, USA) were first compared before Mg^2+^ potentiation experiments were conducted. If the steady-state current in the presence of CPA was larger than 5% of the current in the absence of CPA, the tight seal was considered not to be optimal and the patch was discarded.

### 4.3. Data Analysis

Experimental data were analyzed using Clampfit software (Molecular Devices, San Jose, California, USA), OriginPro 2018 software (OriginLab, Co., Northampton, Massachusetts, USA) and in-house script in Python 3.7 (Python 3.7, first released in 27 June 2018, was downloaded from http://www.python.org); the Python script was used to find the peak current (I_peak_) in recordings. In brief, I_peak_ was detected with a moving average based algorithm, which alleviates a bias detection of the peak from noise of the recording trace. All averaged data are presented as mean ± SEM. The one-way ANOVA followed by Bonferroni’s multiple comparisons was used for hypothesis testing in statistical comparisons. Symbols * and ** indicate the statistically significance levels of 0.05 and 0.005, respectively.

## Figures and Tables

**Figure 1 ijms-22-02209-f001:**
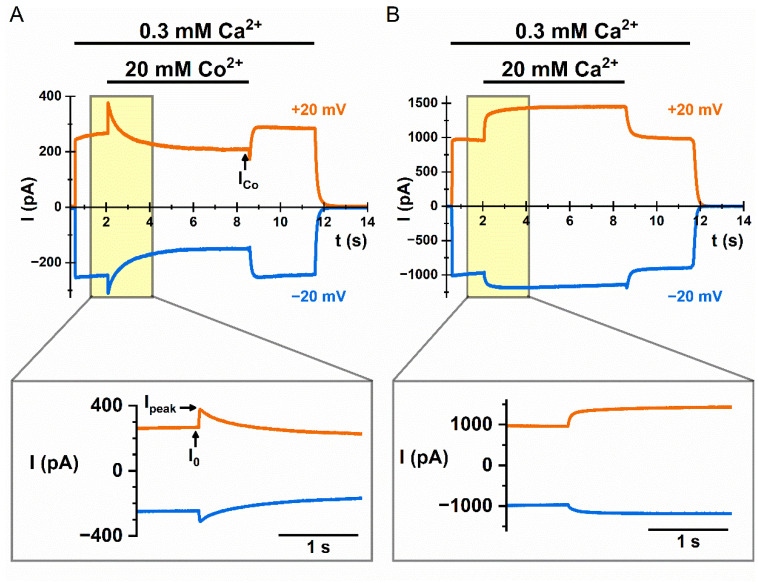
Potentiation of the TMEM16A current by intracellular Co^2+^ and Ca^2^**^+^.** TMEM16A currents were obtained at voltages clamped at −20 mV (blue traces) and +20 mV (orange traces). Currents were activated by 0.3 mM intracellular calcium concentration ([Ca^2+^]_i_). (**A**) Recording traces showing the dual effects, inhibition and potentiation on the wild-type TMEM16A (WT_16A_) current by 20 mM intracellular cobalt concentration ([Co^2+^]_i_). (**B**) Recording traces showing potentiation of WT_16A_ by 20 mM [Ca^2+^]_i_. In the bottom panel of both A and B, an expanded view of the yellow shaded area in the upper panel is depicted to focus on the current potentiation. I_0_ represents the control current before the application of divalent cations, whereas I_peak_ and I_Co_ represent the peak current after the application of Co^2+^ and the quasi-steady-state current at the end of the Co^2+^ application, respectively.

**Figure 2 ijms-22-02209-f002:**
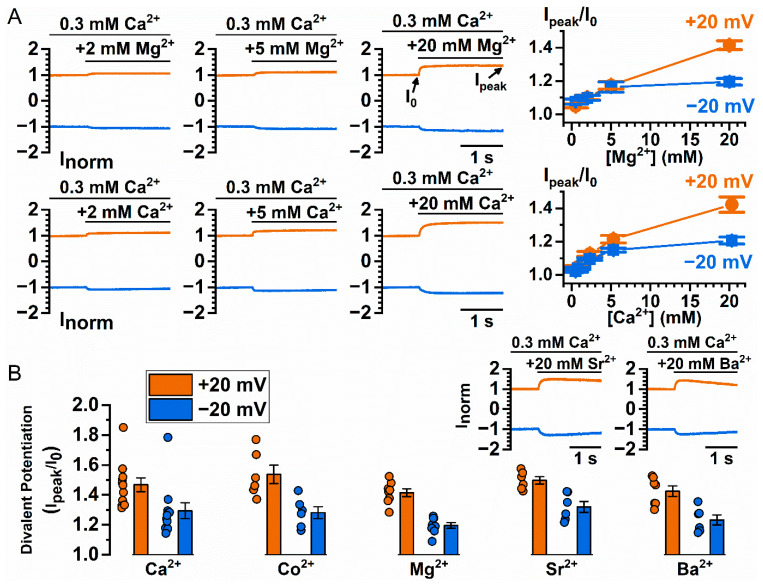
Potentiation of WT_16A_ current by various divalent cations. (**A**) Dose-dependent Mg^2+^ and Ca^2+^ potentiation of the WT_16A_ current. Left three panels show raw recording traces of Mg^2+^ or Ca^2+^ potentiation of WT_16A_ current. The currents were normalized to the current immediately before the application of mM intracellular concentration of magnesium ([Mg^2+^]) or [Ca^2+^]. Right panel shows averaged potentiation of WT_16A_ current as a function of [Mg^2+^] (*n* = 5–8) or [Ca^2+^] (*n* = 6–7). (**B**) Averaged potentiation of 0.3 mM [Ca^2+^]_i_-induced WT_16A_ current by 20 mM Ca^2+^, Co^2+^, Mg^2+^, Sr^2+^ or Ba^2+^ (*n* = 6–11). Insets show raw recording traces of Sr^2+^ or Ba^2+^ potentiation of WT_16A_ current.

**Figure 3 ijms-22-02209-f003:**
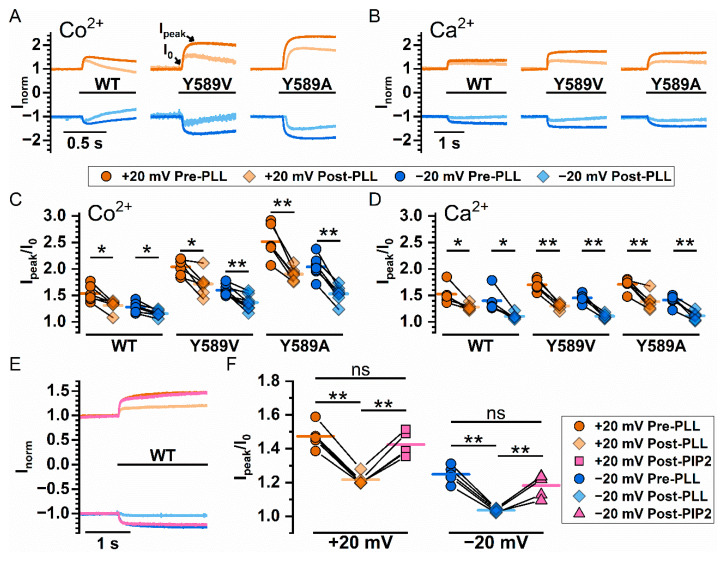
Manipulating Co^2+^ and Ca^2+^ potentiation of WT_16A_, Y589V_16A_, and Y589A_16A_ by intracellular reagents that affect membrane phospholipids. (**A**,**B**) Representative recordings showing Co^2+^ and Ca^2+^ potentiation, respectively, before and after treating the patch with poly-l-lysine (PLL, 0.3 mg/mL) for 5 sec. All currents were induced by 0.3 mM [Ca^2+^]_i_ and potentiated with an additional 20 mM Co^2+^ or Ca^2+^ (black bars underneath phenotype labels). (**C**,**D**) Degree of Co^2+^ and Ca^2+^ potentiation, respectively, before and after poly-l-lysine treatment. Orange (+20 mV) and blue (−20 mV) circles are the potentiation before poly-l-lysine treatment while light orange (+20 mV) and light blue (−20 mV) diamonds are the potentiation after poly-l-lysine. (**E**,**F**) Effects of PIP2 for reversing the effect of poly-l-lysine on the Ca^2+^-induced potentiation of the WT_16A_ current. Degree of Ca^2+^ potentiation was measured before poly-l-lysine treatment, after poly-l-lysine treatment, and after PIP2 treatment. In (**C**,**D**,**F**), data points from the same patch are connected by solid lines, and colored horizontal line segments represent the mean of the data set. ns *p* > 0.05; * *p* < 0.05; ** *p* < 0.005 by one-way ANOVA followed by Bonferroni’s multiple comparisons.

**Figure 4 ijms-22-02209-f004:**
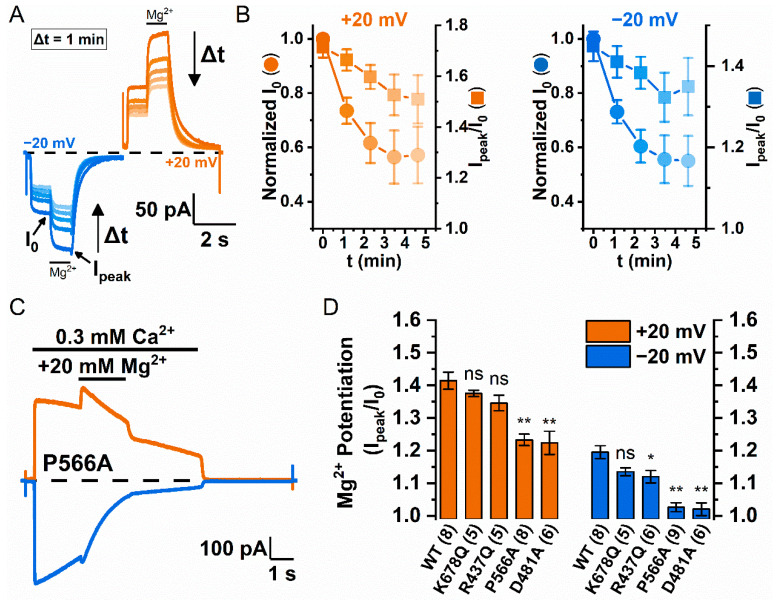
Involvement of Phospholipids in divalent cation-induced potentiation. (**A**) Representative recording traces are depicted to illustrate the decrease in Mg^2+^ potentiation of Y589A_16A_ after channel rundown. For every minute, current was elicited with 100 µM Ca^2+^ (a concentration chosen for reducing the speed of rundown), and 20 mM Mg^2+^ were applied subsequently for 1 s (black bar above traces). (**B**) Reduction of Mg^2+^ potentiation (I_peak_/I_0_) over time at +20 mV (left panel, orange) and −20 mV (right panel, blue) (*n* = 6–9). The rundown of the control current (I_0_ normalized to the I_0_ of the trace at t = 0 min) is shown by circles, whereas the reduction of the Mg^2+^ potentiation is shown by squares. (**C**) Mg^2+^ potentiation of a PIP2 binding-site mutant P566A_16A_ at ±20 mV. The P566A_16A_ current was easier to run down and the degree of potentiation was also smaller compared to that in WT_16A_. (**D**) Comparing Mg^2+^ potentiation at ±20 mV between WT_16A_ (replotted from Figure 1) and four mutants (numbers below each column are the number of patches). ns *p* > 0.05; * *p* < 0.05; and ** *p* < 0.005 by one-way ANOVA followed by Bonferroni’s multiple comparisons.

**Figure 5 ijms-22-02209-f005:**
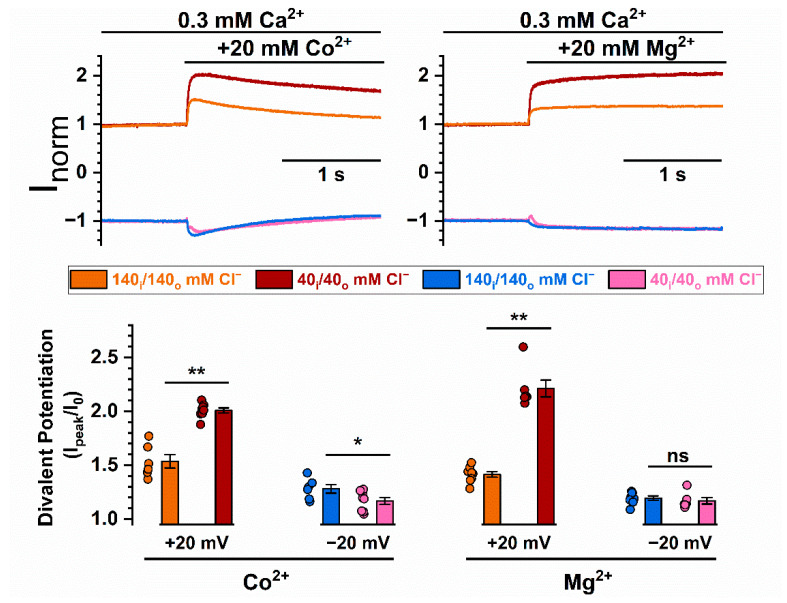
Enhanced divalent cation potentiation of WT_16A_ in low ionic strength solutions. (Top) Recording traces comparing the Co^2+^ or Mg^2+^ potentiation of WT_16A_ between conditions of symmetrical 140 mM and 40 mM NaCl. The 40 mM NaCl solution also contains 100 mM D-mannitol. All currents were induced by 0.3 mM [Ca^2+^]_i_. (Bottom) Bar graph summarizing Co^2+^ and Mg^2+^ potentiation under symmetrical 140 mM or 40 mM NaCl (*n* = 6–8). ns *p* > 0.05; * *p* < 0.05; ** *p* < 0.005 by one-way ANOVA followed by Bonferroni’s multiple comparisons.

**Figure 6 ijms-22-02209-f006:**
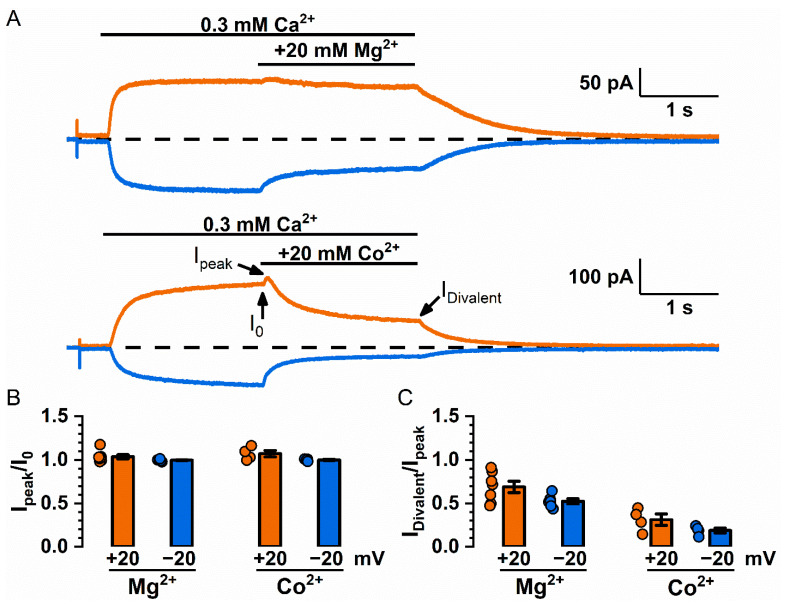
Potentiation and inhibition of the Q559W_16F_ current by Mg^2+^ and Co^2+^. (**A**) Representative recording traces for the Mg^2+^ and Co^2+^ effects on the Q559W_16F_ current (induced by 0.3 mM [Ca^2+^]_i_). (**B**) Averaged potentiation (I_peak_/I_0_) of Mg^2+^ and Co^2+^ on the Q559W_16F_ current. (**C**) Averaged inhibition (I_Divalent_/I_peak_, i.e., I_Mg_/I_peak_ or I_Co_/I_peak_) of Mg^2+^ and Co^2+^ on the Q559W_16F_ current.

**Figure 7 ijms-22-02209-f007:**
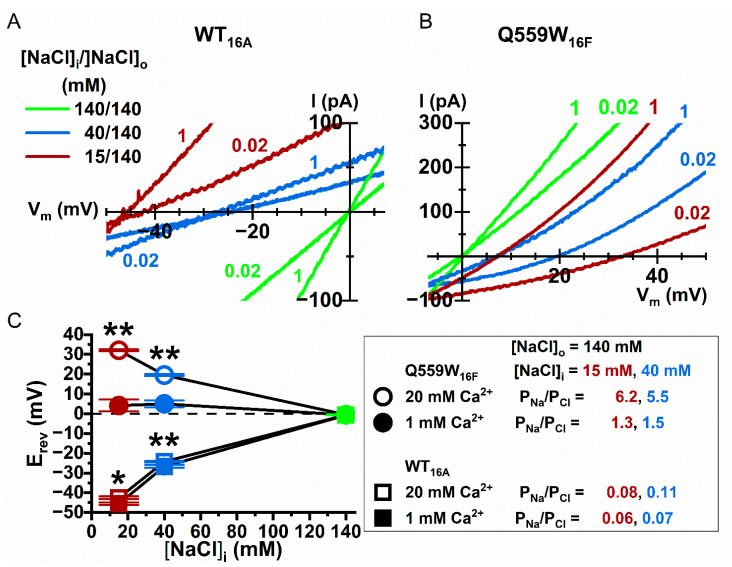
Intracellular Ca^2+^ effect on the Na^+^ versus Cl^-^ permeability ratios (P_Na_/P_Cl_) of TMEM16 currents. Representative I–V curves for (**A**) WT_16A_ and (**B**) Q559W_16F_ under asymmetrical [NaCl]. [NaCl]_o_ = 140 mM in all recordings, whereas [NaCl]_i_ was 140 mM (green), 40 mM (blue) and 15 mM (red), respectively. The reduced [NaCl]_i_ in the 40 and 15 mM [NaCl]_i_ solutions was replaced with D-mannitol. Currents were elicited with either 0.02 mM or 1 mM [Ca^2+^]_i_, indicated by the colored numbers next to each curve. (**C**) Summary of reverse potential (E_rev_) measured under asymmetrical [NaCl] for WT_16A_ (squares) and Q559W_16F_ (circles). The P_Na_/P_Cl_ ratios (calculated based on the Goldman–Hodgkin–Katz equation, see equation 1 in Materials and Methods) of WT_16A_ and Q559W_16F_ are shown in the box on the right and in Table 1 (*n* = 5–15). * *p* < 0.05; ** *p* < 0.005 by one-way ANOVA followed by Bonferroni’s multiple comparisons.

**Figure 8 ijms-22-02209-f008:**
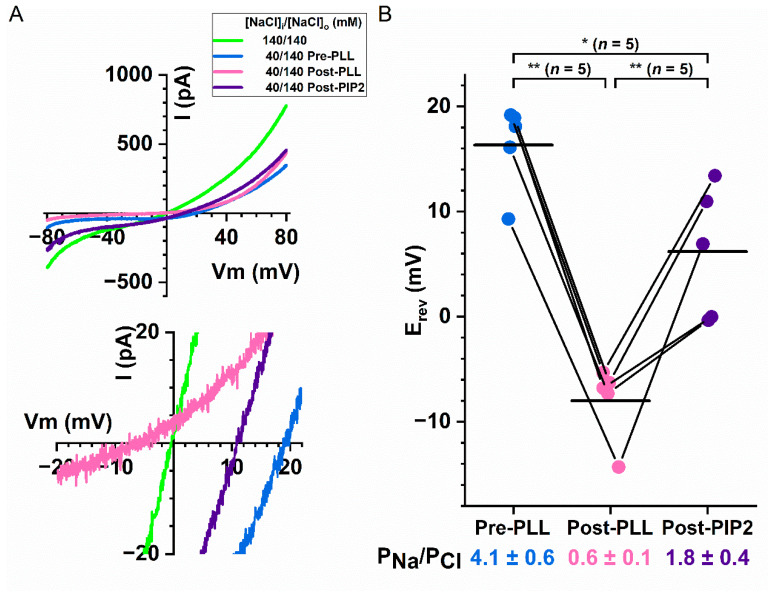
Effects of manipulating membrane phospholipids on the P_Na_/P_Cl_ ratio of Q559W_16F_. All experiments were performed with 20 µM [Ca^2+^]_i_. [NaCl]_o_ = 140 mM in all experiments. (**A**) Representative I–V curves of Q559W_16F_ in various intracellular solutions; −80 mV to +80 mV are shown in the top panel, whereas the expanded traces near reversal potentials are depicted at the bottom. Experiments were first performed in 140 mM [NaCl]_i_ (green traces), followed by experiments in 40 mM [NaCl]_i_ before (blue traces) and after (pink traces) 0.3 mg/mL intracellular poly-l-lysine treatment for 5 sec. Finally, the I–V curve was obtained after the patch was intracellularly treated with 20 µM PIP2 for 1 min (purple traces). (**B**) Altering the P_Na_/P_Cl_ ratio of Q559W_16F_ after treating membrane patches with intracellular poly-l-lysine or PIP2. Results are from experiments like those shown in (A) and data points from the same patch are connected by line segments. Horizontal lines depict the averaged reversal potentials from individual data set. The mean P_Na_/P_Cl_ ratios (± SEM) at the bottom of the plot were calculated according to equation 1. * *p* < 0.05; ** *p* < 0.005 by one-way ANOVA followed by Bonferroni’s multiple comparisons.

**Figure 9 ijms-22-02209-f009:**
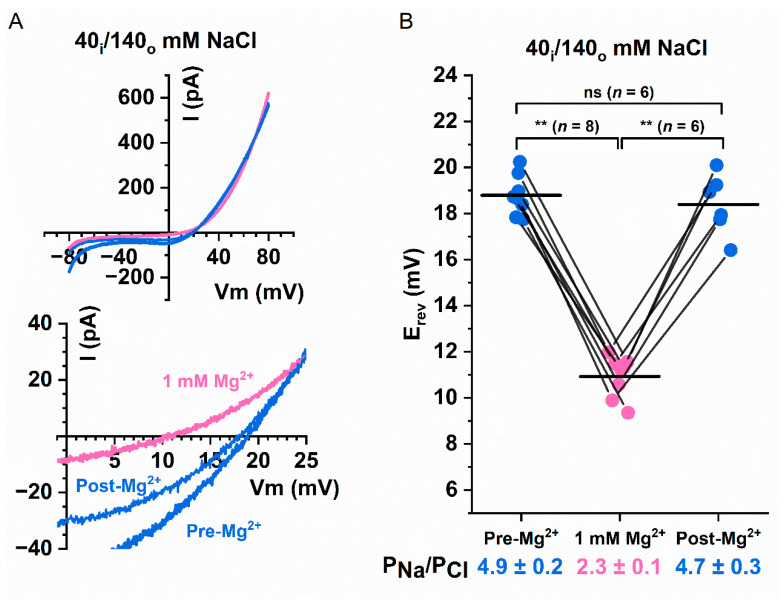
Effects of [Mg^2+^]_i_ on the P_Na_/P_Cl_ ratio of Q559W_16F_. (**A**) Representative I–V curves of Q559W_16F_ under asymmetrical [NaCl] in the presence and absence of 1 mM Mg^2+^. Currents were activated by 20 µM [Ca^2+^]_i_. Bottom panel shows the same I–V curves expanded around E_rev_. (**B**) Paired data showing the values of E_rev_ in the presence (pink circles) of 1 mM Mg^2+^, and the values of E_rev_ before Mg^2+^ wash-in and after Mg^2+^ wash-out (blue circles). Horizontal black lines indicate the mean value for each data set. Calculated P_Na_/P_Cl_ ratios are shown at the bottom of the plot. ns *p* > 0.05; ** *p* < 0.005 by one-way ANOVA followed by Bonferroni’s multiple comparisons.

**Figure 10 ijms-22-02209-f010:**
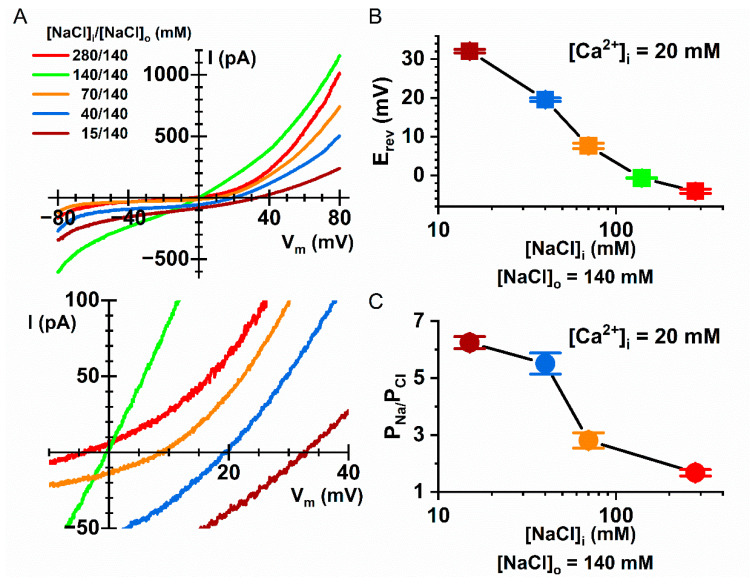
Dependence of the P_Na_/P_Cl_ ratio of Q559W_16F_ on [NaCl]_i_. (**A**) Representative I–V curves for Q559W_16F_ in various [NaCl]_i_ (from 15 to 280 mM). In all recordings, [NaCl]_o_ = 140 mM and the currents were activated by 20 µM [Ca^2+^]_i_. The same I–V curves expanded around the E_rev_ are shown in the bottom panel. (**B**) Averaged E_rev_ as a function of [NaCl]_i_ from the recordings like those shown in A. (**C**) P_Na_/P_Cl_ ratio as a function of [NaCl]_i_. Note the reduction of the P_Na_/P_Cl_ ratio as [NaCl]_i_ increases (*n* = 6–22).

**Figure 11 ijms-22-02209-f011:**
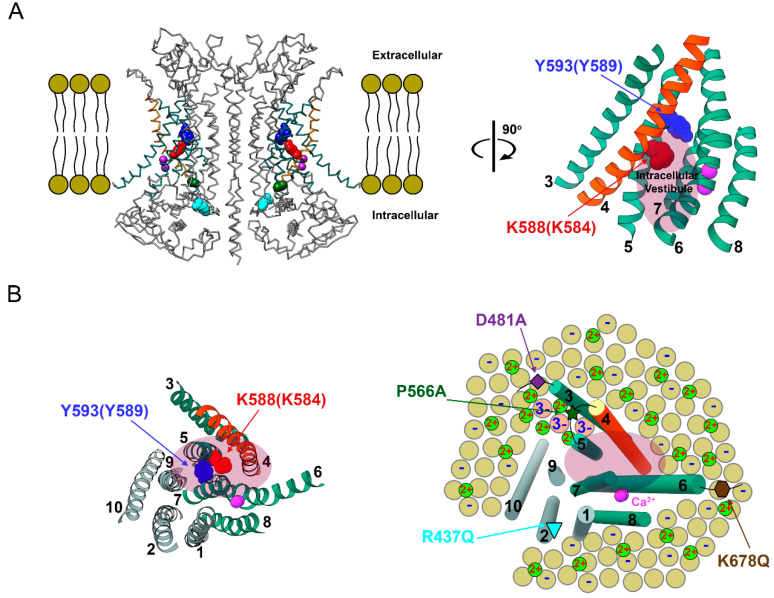
Illustration of divalent cation effects on TMEM16 molecules. (**A**) High resolution structure of the “ac” alternatively spliced variant of TMEM16A (left, PDB:5OYB). The six transmembrane helices (helices 3–8) of a single subunit forming the ion-conduction pathway are rotated 90° clockwise along the axis perpendicular to the cell membrane (right). Helix 4 is colored in orange, whereas all other helices are colored in green. Residue K584 of the alternatively spliced “a” variant of TMEM16A (used in our experiments) and residue Q559 of TMEM16F corresponds to residue K588 of the TMEM16A “ac” variant (colored in red). Y589 of the TMEM16A “a” variant mentioned in the text corresponds to Y593 in the TMEM16A “ac” variant (colored in blue). Light purple oval roughly depicts the intracellular pore vestibule. (**B**) Intracellular view perpendicular to the cell membrane of a single subunit. (Left) all transmembrane helices (with helix numbers) are shown. (Right) Cartoon model of the six pore-forming helices depicted as cylinders (same orientation as that in the left panel). Intracellular leaflet of cell membranes contains negatively charged phospholipids (yellow circles labeled with “−”) as well as neutral ones (yellow circles without “−”). PIP2 molecules are depicted as salmon-color circles with “3−”. The Ca^2+^ ions at the activation sites are colored in pink. Monovalent (not shown), divalent (depicted as small green circles) or even multivalent cations (not shown) can bind to phospholipid head groups, consequently decreasing the negative potential from phospholipids. Four mutations associated with PIP2 regulations of TMEM16A studied in this paper are shown. Notice that residues P556 (dark green star) and D481 (purple diamond) appear to be closer to the intracellular pore vestibule than residues R437 (cyan down triangle) and K678 (brown hexagon).

**Table 1 ijms-22-02209-t001:** Calculated P_Na_/_PCl_ of WT_16A_ and Q559W_16F._

[NaCl]_I_ ^ǂ^ (mM)	[Ca^2+^]_i_ (mM)	P_Na_/P_Cl_	
		WT_16A_	Q559W_16F_
15	0.02	0.08 ± 0.005	6.2 ± 0.2
	1	0.06 ± 0.004	1.3 ± 0.2
40	0.02	0.11 ± 0.006	5.5 ± 0.4
	1	0.07 ± 0.01	1.5 ± 0.2
70	0.02	--	2.8 ± 0.3
	1	--	--
280	0.02	--	1.7 ± 0.1
	1	--	--

ǂ Does not include extraneous Na^+^ from NaOH for clarity. However, the extraneous Na^+^ was included for the calculation of the P_Na_/P_Cl_ ratio.

## Data Availability

The data presented in this study are available on request from the corresponding author. The data are not publicly available due to privacy.

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
