# Peer review of "Divalent Cation Modulation of Ion Permeation in TMEM16 Proteins"

_ijms, 2021, doi:10.3390/ijms22042209_

Round 1

Reviewer 1 Report

This manuscript by Nguyen et al. investigated how divalent cations modulated ion permeation of TMEM16 proteins and its associated molecular mechanism. In general the experiments were extensively carried out. The authors conclude that divalent cations could increase local concentrations of permeant ions via a change of pore electrostatic potential, through a potential phospholipid head groups in or near the pore, well supported by the data presented. I have only several minor comments.

  1. As the authors acknowledged, the physiological relevance of such study is unclear. The concentrations of all ions applied to activate or modulate the channel were unusually high, far beyond the typical physiological concentration. The authors need to clarify and discuss in more details the possibilities how this activation and modulation takes place in pathological condition or how such study benefit other research such as drug discovery.
  2. In Figure 1, Ico indication should be clarified. And an appropriate and meaningful statistics figure should be accompanied with Figure 1 to conclude these findings here, e.g. the time course of modulation by the ions etc.
  3. The colour used throughout all figures is quite misleading. The authors should strive to make the colour representing different condition as consistent as possible. It is quite misleading in the current version and sometimes quite misleading now.

Reviewer 2 Report

This manuscript clarified the biophysical mechanism of divalent cation modulation on the ion permeation of TMEM16A and 16F. Overall, this manuscript is written  well, and the experimental techniques and strategies are reliable. There are a few concerns that need to be addressed.

Major concerns:

1. In Figure 3F and Figure 4D, the results were analyzed by multiple comparisons but not paired t-test.

2. ‘References’ should be numbered, according to ‘Instruction for Authors’.

3. Section 5 ‘Conclusions’ must be added.

Minor concerns:

1. Page 4, line 3 from the bottom: Approximately 40 - 60% increase in outward currents at +20 mV and 20 - 40% increase in inward currents at -20 mV compared with the control current (activated by 0.3 mM [Ca2+]i) …

2. Page 8, line 6: located further away

3. Please amend EGTA in ‘Abbreviations’.

Reviewer 3 Report

This paper describes the modulation of TMEM16 by divalent cations in the cell.

Although the physiological relevance of this phenomenon is not clear as the authors mention, the results obtained in this paper are important for understanding the mechanism of TMEM16 regulation in cells.

Round 2

Reviewer 2 Report

The authors adequately responded to the reviewer's comments. I have no more concerns.